# The Safety and Efficacy of Vela Laser En-Bloc Endoscopic Resection versus Conventional Transurethral Resection of Bladder Tumor—A Single Center Experience

**DOI:** 10.3390/jcm11175233

**Published:** 2022-09-05

**Authors:** Che-Wei Chang, Tsz-Yi Tang, Jiun-Hung Geng, Jhen-Hao Jhan, Hsun-Shuan Wang, Jung-Tsung Shen, Yung-Chin Lee

**Affiliations:** 1Department of Urology, Kaohsiung Municipal Siaogang Hospital, Kaohsiung Medical University, Kaohsiung 81267, Taiwan; 2Department of Urology, Kaohsiung Medical University Hospital, Kaohsiung Medical University, Kaohsiung 80708, Taiwan; 3Department of Urology, School of Medicine, College of Medicine, Kaohsiung Medical University, Kaohsiung 80708, Taiwan; 4Graduate Institute of Clinical Medicine, College of Medicine, Kaohsiung Medical University, Kaohsiung 80708, Taiwan

**Keywords:** bladder tumor, laser resection, en-bloc, bipolar, monopolar

## Abstract

(1) Background: The current gold standard treatment of bladder cancer is conventional transurethral resection of the bladder tumor (CTURBT) using monopolar or bipolar resectoscopes. Laser en-bloc resection of the bladder tumor (LERBT) could achieve a higher quality of the specimen, reduce perioperative complications, and decrease the recurrence rate. Here, we compare the efficacy and safety of en-bloc Vela laser resection versus the conventional monopolar/bipolar resection; (2) Methods: A total of 100 clinically cT1-2 patients with bladder cancer were retrospectively reviewed in this study. Among these patients, 50 patients received LERBT, and 50 patients received CTURBT. The baseline characteristics, operation variables, and clinical outcomes were collected. The primary performance was the presence of muscle layer in the specimen. Perioperative complications and recurrence-free survival (RFS) were also compared. Independent *t*-test, Chi-square test, Kaplan–Meier curves, and the Cox-regression model were used in the analysis; (3) Results: The median age of the patients in the laser and resectoscope groups was 69.2 and 68.0 years old, respectively. The statistical difference in the presence of the detrusor muscle was 92.0% in the laser group and 70.0% in the CTURBT group (*p* = 0.005). A lower incidence of bladder perforation (*p* = 0.041) and major surgical complications (*p* = 0.046) in the LEBRT group was observed. We found no differences in operation duration, catheterization time, and hospitalization time after adjustment. Additionally, there was no statistical difference in RFS after a median follow-up time of 25 months; (4) Conclusions: Endoscopic laser en-bloc resection of bladder tumor with Vela laser is an effective method with higher muscle inclusion rate and fewer complications.

## 1. Introduction

Bladder cancer ranks as the ninth most commonly diagnosed cancer worldwide [1]. In Taiwan, bladder cancer is the second most common urological cancer [2]. The most common risk factors are tobacco smoking, end-stage renal diseases, and chemical exposures such as arsenic [3,4,5]. Around 75% of newly diagnosed bladder cancers are confined to the mucosa (pTa) or submucosa (pT1) [6].

The initial treatment for noninvasive bladder cancer is through a proper and thorough transurethral resection (TUR) combined with intravesical therapy according to current guidelines [7]. All guidelines agree that an incomplete resection or the absence of the detrusor requires repeated TUR sampling within 1–6 weeks from the initial surgery [8,9]. Thus, optimized resection is essential for risk assessment and may improve the outcome [10].

Conventional transurethral resection of the bladder tumor (CTURBT) with a monopolar or bipolar resectoscope obtains an inferior quality of specimen than laser en-bloc resection of the bladder tumor (LERBT) based on a previous meta-analysis study [11]. More perioperative complications such as bladder perforation, obturator nerve jerk, and bleeding were documented during conventional resection compared to en-bloc resection [12].

Laser en-bloc resection of the bladder tumor (LERBT) is a complete tumor resection technique. LERBT improves the quality of resection, reduces perioperative complications, and has the potential to improve recurrence rates [13,14,15,16]. Several laser types for ERBT have been described, such as holmium, thulium, and potassium-titanyl-phosphate. However, there has been limited evidence about the Vela laser for resection of bladder tumors. Moreover, a previous retrospective cohort study using a Vela laser did not analyze the presence of muscle in specimens [17,18].

Here, we aimed to evaluate the safety and efficacy of Vela laser en-bloc resection of the bladder tumor versus CTURBT for patients with clinical stage cT1 and cT2 bladder tumor.

## 2. Materials and Methods

### 2.1. Data Collection

From September 2018 to April 2022, the medical records of patients with pathologically proven urothelial carcinoma at a single center were reviewed. The inclusion criteria were patients who had bladder urothelial cancer with clinical stage cT1 to cT2. All of them must have had pathological tissue removed by using a laser or traditional mono- or bipolar resectoscope. Subjects were excluded for those with other histological variants, lacking perioperative and oncological variables, and follow-up time less than 3 months. No experiments on humans were conducted, and thus informed consent and guidelines were not required. This study was approved by the ethical review board ([KMUHIRB-E(I)-2021-0402]). The requirement for consent and guidelines was waived by institutional review boards.

All the operations were performed according to a standardized procedure. For the LERBT group, a Vela laser (Vela™ XL, Boston Scientific, Marlborough, MA, USA) en-bloc resection was performed with 600 nm fiber and 30 W energy. As for CTURBT group, the resection was performed using a piece-by-piece method using monopolar or bipolar cauterization. The resection was performed by incising 0.2–0.5 cm away from the visible tumor edge and down to the muscular layer in both groups. After the operation, the catheter could be removed if no obvious gross hematuria was noted. Patients would be discharged on the second day if no urinary retention was observed. Regarding the intravesical chemotherapies, 6–8 cycles of instillation of mitomycin C 30 mg were performed in selected patients.

Clinical variables including age, sex, hypertension (HTN), diabetes mellitus (DM), chronic kidney disease (CKD), smoking status, previous bladder cancer history, previous upper urinary tract urothelial carcinoma (UTUC) history, tumor size, and tumor grade were collected. CKD was defined as glomerular filtration rate (GFR) < 60 mL/min/1.73 m^2^. Perioperative variables including bladder perforation rate, detrusor muscle inclusion rate, operation time, catheterization time, and hospital stay were also collected. Cancer recurrence-free survival was recorded until April 2022.

### 2.2. Study Outcome

The primary outcome of this study was the detrusor muscle inclusion rate between LERBT and CTURBT groups. The secondary outcome was the differences in perioperative factors, complication rate, and recurrence-free survival between these two groups, which was defined as bladder cancer recurrence identified based on an image study or cystoscopy.

### 2.3. Statistical Analyses

The statistical differences between LERBT and CTURBT were assessed using Pearson’s chi-square test for categorical variables and student t-test for numeric variables. For, those who lost follow-up, they were excluded from the recurrence-free survival analysis. Recurrence-free survival was analyzed using the Cox-regression model. Adjusted variables included age, sex, previous bladder tumor, previous UTUC, tumor size, tumor number, tumor grade, tumor stage, intravesical chemotherapy, adjuvant chemotherapy, and detrusor muscle inclusion rate. Kaplan–Meier survival curves were plotted according to the Cox-regression model. All analyses were carried out using SPSS 19.0 (IBM Corp, Armonk, NY, USA). Statistical tests with *p*-values lower than 0.05 were considered to be significant.

## 3. Results

In total, 100 patients enrolled from September 2018 to April 2022; 50 receiving LERBT and 50 receiving CTURBT were retrospectively analyzed. The mean ages among the LERBT and CTURBT groups were 69.2 ± 9.9 and 68.0 ± 12.3 years, respectively. No significant differences existed in comorbidities such as HTN, DM, and CKD. Additionally, there are no differences in personal history, including smoking, concomitant UTUC, previous bladder tumor history, previous UTUC history, and tumor number. No significant differences were observed in pathological stage, tumor grade, and adjuvant chemotherapy (Table 1).

The intra- and postoperative characteristics between the two groups are listed in (Table 2). No statistical difference was seen in operation time, hospitalization time, and catheterization time. The detrusor muscle inclusion rate was higher in the LERBT group than that of the CTURBT (92.0% vs. 70.0%, *p* = 0.005). All laser en-bloc resections of cT2 tumor samples showed the presence of the muscle layer (100%), compared to the CTURBT group’s 77.8% (*p* = 0.038). In addition, there was a significant difference in major surgical complication rates between LERBT and CTURBT (0% vs. 4.0%, *p* = 0.046). There were no cases of bladder perforation in the LERBT group, but four (8%) patients experienced bladder perforation in the CTURBT group. No obturator reflex jerk was recorded in the medical records.

The mean follow-up time was around 25 months in both groups. No differences were noted in terms of RFS, though the LERBT group had a longer recurrence-free time than the CTURBT group (12.01 months vs. 10.54 months). The RFS curve was plotted and showed no obvious differences between the two groups (Figure 1). The Cox-regression model reported hazard ratios are presented in Table 3. No statistical differences were observed in other adjusted variables.

## 4. Discussion

The goal of the transurethral resection of nonmuscle invasive tumor is to completely eradicate visible bladder tumors, which is crucial for initial diagnosis and treatment [19]. However, CTURBT has many drawbacks, such as obturator nerve jerk and deeper thermal damage, which could lead to the absence of the muscle layer in the specimen and positive margins [20]. Most importantly, this “incise-and-scatter” technique could produce exfoliated tumor cells which could lead to recurrence [21]. Furthermore, a meta-analysis showed 51% persistent diseases and an 8% risk of understaging in T1 bladder tumors [10]. Therefore, repeated TURBT is indicated for incomplete resection, finding of T1 disease, and absence of detrusor muscle [8].

Endoscopic laser procedure has been widely used in various surgeries, including urological operations [22]. To overcome the drawbacks of CTURBT, various techniques have been reported, including Holmium yttrium-aluminum-garnet (YAG), Thulium YAG, Kalium-titanyl-phosphate (KTP) laser, and water jet [19]. Recent advancement of LERBT has been reported to be effective and safe in nonmuscle invasive bladder cancer [16,18,23,24]. However, most studies enrolled patients with nonmuscle invasive bladder cancer, including cTa, cT1, and cTis. In this study, we included cT2 cases because of the ambiguity of cT1 and cT2 based on CT.

Vela laser XL comprises a 1.94 µm thulium wavelength and a shallow penetration depth of 0.2 mm. It has a continuous energy output that guarantees better hemostasis and cutting efficiency. Furthermore, it confers a more precise cut without thermal damage compared to CTURBT. Thus, a good quality of specimens could be obtained without damaging the tissue. Additionally, it reduces intraoperative complications such as bladder perforation and obturator nerve jerk [18]. As for other laser, Holmium has a 2100 nm wavelength with a penetration depth of 0.5 mm, and KTP laser has a 532 nm wavelength and a penetration depth of 0.1 mm. Because the depth of urothelium plus submucosa is between 0.46~3.07 mm [25], the Vela laser, with a 0.2 mm penetration energy, is more suitable for incising this layer.

In the present study, there was no statistical difference in the operation time between the two groups. It is compatible with previous studies that operation time could be altered by tumor number, size, and site, and by different resection methods [16,18,26]. No significant differences in catheterization time and hospitalization time were observed between both groups. These results are similar to previous studies, which recorded catheterization time ranging from 1.9 to 5.2 days in the LERBT groups and 2.8 to 5.6 days in the CTURBT groups. Hospitalization duration ranges from 2.3 to 5.8 days and 3.1–6.4 days, respectively [14].

A multicenter study revealed that both electrical (monopolar or bipolar) ERBT and LERBT groups had a similar result of low Clavien–Dindo grade complications. However, the LERBT group had a lower conversion rate to CTURBT, which is probably implicating that LERBT is preferable for complete resection of larger tumors due to its precise cutting line and better operation vision [12]. In the current study, there was a significantly lower incidence of bladder perforation and major surgical complications in the LERBT group compared with the CTURBT group. A meta-analysis study reported a significantly lower rate of bladder perforation in patients treated with LERBT (OR, 0.17; 95% Cl, 0.09 to 0.35; *p* < 0.00001) than those who underwent CTURBT [11]. The incidence of bladder perforation reported in the LERBT and CTURBT groups varied from 0 to 5% and 0.83 to 8.9% [11,18,23,24,26]. Thus, these results suggest that laser en-bloc resection offers a lower complication rate than CTURBT.

The presence of detrusor muscle can be seen as a surrogate marker of qualified specimens. With individual surgeon scorecards, surgeons could improve muscle sampling in the resection of bladder tumors [27]. A good quality of tumor specimens could avoid understaging and further repeated biopsies, minimizing tumor recurrence and progression [28]. However, a lower detrusor muscle sampling rate of CTURBT was observed at around 67~84% [29,30,31,32], necessitating re-resection of the bladder tumor. A previous study using Vela laser did not compare the detrusor muscle inclusion rate with CTURBT [18]. In the present study, the detrusor muscle inclusion rate was 92% in the LEBRT group and 70% in the CTURBT group (*p* = 0.005). More detrusor muscle involvement can avoid early re-resection and prevent these patients from invasive procedures with potential complications and higher financial costs. Regarding the quality of the obtained specimens, no residual tumor was observed in the LERBT group in our study. A multicenter observational study reported by Hurle et al. of 78 patients who underwent LEBRT revealed five (6.41%) residual cancers in patients with high-risk bladder cancer [15].

As for recurrent-free survival, previous studies showed no significant differences between various laser resection techniques and CTURBT [16,18,23,24,26]. We further examined the Cox regression analysis of RFS by adjusting possible confounding factors, including age, sex, previous bladder tumor, previous UTUC, tumor size, tumor number, tumor grade, tumor stage, concomitant UTUC, intravesical chemotherapy, adjuvant chemotherapy, and detrusor muscle inclusion rate. Cox regression analysis after stratification by the detrusor inclusion rate revealed a hazard ratio of 0.054 (95% CI, 0.222 to 1.014; *p* = 0.054), suggesting better survival outcomes in complete resection of bladder tumors with the muscular layer. There are very few studies that investigated long-term oncological outcomes of en-bloc resection. Hurle et al. reported 86% of recurrence-free survival after two years follow-up, and all the en-bloc resection of bladder tumor samples showed the presence of detrusor muscle, and the recurrence rate at the first follow-up cystoscopy (3 months) was 5.4% [13]. In a 5-year follow-up study by Paciotti et al. [33], the monopolar loop was used for ERBT in 74 patients with NMIBC. They found that 57 of 74 patients (77%) were free of recurrence at 5 years follow-up.

In addition to pathological staging, computed tomography urography (CTU) is the most widely used image tool in clinical practice for bladder cancer patients. Recent utilization of multiparametric magnetic resonance imaging (mpMRI) plays a role in the staging of bladder cancer [34,35]. Nevertheless, the high cost of magnetic resonance imaging (MRI) and the complexity of interpretation hinder the detection of the progress of the disease [36]. Therefore, pathological staging is crucial for those who do not receive mpMRI checkup. Based on current findings, we propose that Vela laser is a feasible method for en-bloc resection of nonmuscle invasive bladder tumor and those whose clinical stage is ambiguous between nonmuscle and muscle invasive bladder tumors.

There are several limitations in the present study. First, due to its retrospective nature and some operator preferences in selecting patients, there could be a selection bias. Secondly, there could be an operator bias since the operator’s learning curve would influence the resection quality. Thirdly, due to the limited number of cases and short follow-up time, our results could only demonstrate the feasibility and safety of Vela laser for en-bloc resection of bladder tumors. Further large, multicenter prospective studies with long-term follow-up are required.

## 5. Conclusions

Compared to monopolar/bipolar CTURBT, Vela laser en-bloc resection of bladder tumors permits a higher detrusor muscle inclusion rate. A good quality of specimens can be obtained so that understaging can be avoided. No significant differences were noted in terms of perioperative and oncological outcomes. We suggest that laser en-bloc resection of bladder tumor may be an alternative, feasible method for patients with bladder cancer.

## Figures and Tables

**Figure 1 jcm-11-05233-f001:**
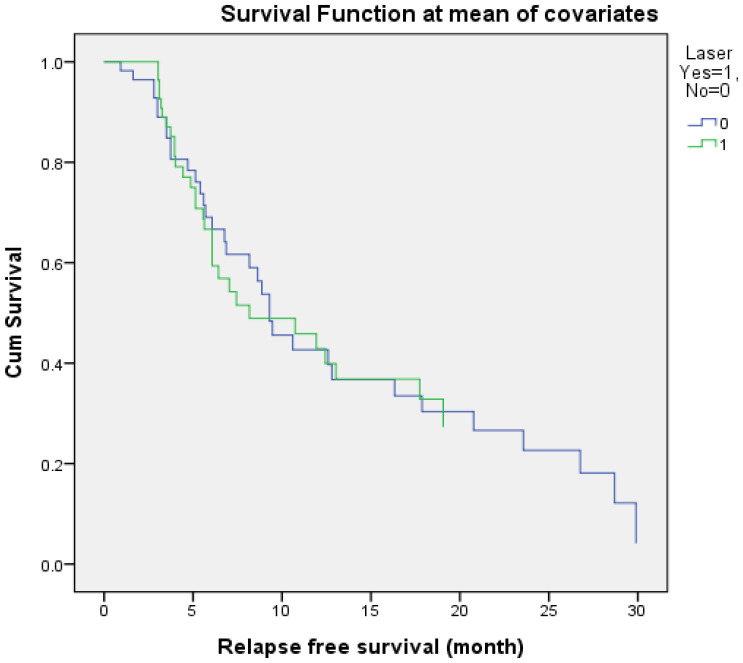
Adjusted Kaplan–Meier estimated recurrence-free survival between laser en-bloc resection of bladder tumor (LERBT) (green) and conventional transurethral resection of bladder tumor (CTURBT) (blue). (Adjusted variables: age, sex, previous bladder tumor, previous upper urinary tract urothelial carcinoma (UTUC), tumor size, tumor number, tumor grade, tumor stage, intravesical chemotherapy, adjuvant chemotherapy, and detrusor muscle inclusion).

**Table 1 jcm-11-05233-t001:** Clinico-pathological characteristics between LEBRT and CTURBT.

Characteristics	LEBRT *n* = 50		CTURBT *n* = 50		*p*-Value
Age ± SD (year)	69.16 ± 9.9		68.02 ± 12.3		0.569
Sex					
Male	39	(78.0%)	33	(34.0%)	0.181
Female	11	(22.0%)	17	(66.0%)	
BMI ± SD (kg/m^2^)	25.0 ± 3.5		24.4 ± 3.5		0.66
ECOG					
1	49	(98%)	48	(96%)	0.603
2	1	(2%)	1	(2%)	
HTN	29	(58%)	23	(46%)	0.230
DM	16	(32.0%)	11	(22%)	0.260
CKD	26	(52%)	21	(42%)	0.316
Smoker	14	(28%)	8	(16%)	0.148
Concomitant UTUC	3	(6%)	3	(6%)	1
Previous bladder tumor	11	(22%)	17	(34%)	0.181
Previous UTUC	3	(6%)	0	(0%)	0.079
Tumor number					
single	33	(66%)	30	(60%)	0.530
multiple	17	(34%)	20	(40%)	
Tumor size					
Mean size ± SD (cm)	1.2 ± 0.64		0.96 ± 0.63		0.852
Tumor grade					
low grade	14	(28%)	17	(34%)	0.517
high grade	36	(72%)	33	(66%)	
Clinical stage					
cT1	39	(78%)	41	(82%)	0.617
cT2	11	(22%)	9	(19%)	
Pathologic staging					
stage pTa	36	(72%)	39	(78%)	0.787
stage pT1	9	(18%)	7	(14%)	
stage pT2	5	(10%)	4	(8%)	
intravesical chemotherapy	35	(70%)	40	(80%)	0.243
Adjuvant chemotherapy	5	(10%)	4	(8%)	0.727

CTURBT, conventional transurethral resection of the bladder tumor; LERBT, Laser en-bloc resection of the bladder tumor; SD, standard deviation; BMI, body mass index; HTN, hypertension; DM, diabetes mellitus, CKD, chronic kidney disease; UTUC, upper urinary tract urothelial carcinoma.

**Table 2 jcm-11-05233-t002:** Intra- or postoperative characteristics between LEBRT and CTURBT.

Characteristics	LEBRT *n* = 50	CTURBT *n* = 50	*p*-Value
Operation duration (min)	47.86 ± 2.6	44.48 ± 3.2	0.441
Hospitalization time (day)	4.1	3.6	0.989
Catheterization time (day)	3.06	3.52	0.197
Detrusor muscle inclusion	46 (92.0%)	35 (70.0%)	0.005
Detrusor muscle inclusion (cT2)	11/11 (100%)	7/9 (77.8%)	0.038
Bladder perforation rate	0	4 (8%)	0.041
Major surgical complications	0	2 (4.0%) *	0.046
Mean follow-up time (month)	23.5	26.5	0.207
Recurrence-free survival (month)	12.01	10.54	0.259

* one bleeding complication required blood transfusion, one sepsis.

**Table 3 jcm-11-05233-t003:** Cox-regression analysis of recurrence-free-survival.

Characteristics	Hazard Ratio (CI)		*p*-Value
Age	0.99	(0.964 to 1.012)	0.309
Sex	0.98	(0.518 to 1.862)	0.957
Previous bladder tumor	0.79	(0.222 to 1.014)	0.483
Previous UTUC	0.91	(0.176 to 4.683)	0.909
Tumor size	1.16	(0.759 to 1.763)	0.499
Tumor number	1.18	(0.925 to 1.496)	0.184
Tumor grade	0.83	(0.424 to 1.642)	0.600
Tumor stage			0.653
Intravesical chemotherapy	0.28	(0.025 to 3.113)	0.298
Adjuvant chemotherapy	0.42	(0.131 to 1.317)	0.136
Detrusor muscle inclusion	0.48	(0.402 to 1.539)	0.054

## Data Availability

Not applicable.

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
