# Peer review of "The Safety and Efficacy of Vela Laser En-Bloc Endoscopic Resection versus Conventional Transurethral Resection of Bladder Tumor—A Single Center Experience"

_jcm, 2022, doi:10.3390/jcm11175233_

Round 1

Reviewer 1 Report (Previous Reviewer 1)

Thank to the authors for the revisions done after the first review.

However, I still have some comments.

1. Abstract

It is stated in the results that there are no differences in the complication rates afte adjustment. In contrast, the authors conclude that there are fewer complications.

Introduction

I do not understand the last sentence (line 19).

The safety and efficacy will be evaluated = ok...

...but the ability to differentiate NMIBC and MIBC. This is another study aim? Where is the evaluation and outcome of this? To compare with what? MRI? CTURB?

Materials and Methods

Data collection  ok

Study outcome

Primary outcome is clear

Secondary outcome: differences in perioperative factors and complication rate are missing

results

line 25-26 - there are no differences in RFS between laser and CTURB (8 months versus 7 months) = I couldnt find these numbers. Maybe 10,5 versus 12 is correct??

Discussion

I do not clearly understand the message on line 34-35. Did you want to say that laser can detect muscle invasion as easily as standard TURB can?? As I can see in table 1, CTURB detected 4 T2 tumors and laser had no problem to detect those tumors too (5).

maybe you can also highlight the fact that with more DM involvement in the speciment more early reTURB can be avoided (as a second early invasive procedure with potential compliations, higher costs etc)

Conclusions

Based on presented retrospective data prospective randomized trials are advocated. 

Author Response

Dear editors and reviewers:

Thank you very much for your comments on our manuscript. We have carefully considered the comments of the reviewers and revised our manuscript accordingly.   Below are our responses to the reviewers' comments.

*Our previous manuscript ID: jcm- 1769402

# Reviewer 1

1.        Abstract: It is stated in the results that there are no differences in the complication rates after adjustment. In contrast, the authors conclude that there are fewer complications.

Thank you for your comments. There is statistical difference in the major surgical complication rates (p=0.046). We’ve revised the statement in the abstract section. (line 28-30 on page 1)  

  1. Introduction: I do not understand the last sentence (line 19). The safety and efficacy will be evaluated = ok......but the ability to differentiate NMIBC and MIBC. This is another study aim? Where is the evaluation and outcome of this? To compare with what? MRI? CTURB?
    Thank you for your comments. we aimed to evaluate the safety and efficacy of Vela laser en-bloc resection of bladder tumor versus CTURBT. We’ve revised this paragraph as your per request. (Introduction: line 15-16 on page 2)
  2. Secondary outcome: differences in perioperative factors and complication rate are missing
    Thank you for your comments. We’ve revised this paragraph. (line 47-48 on page 2)
  3. results: line 25-26 - there are no differences in RFS between laser and CTURB (8 months versus 7 months) = I couldnt find these numbers. Maybe 10,5 versus 12 is correct??
    Thank you for your comments. Yes, LERBT group has longer relapse-free time than CTURBT group (12.01 months vs. 10.54 months). We’ve revised this paragraph. (line 29 on page 3)

  4. Discussion: I do not clearly understand the message on line 34-35. Did you want to say that laser can detect muscle invasion as easily as standard TURB can?? As I can see in table 1, CTURB detected 4 T2 tumors and laser had no problem to detect those tumors too (5).
    Thank you for your comment. Based our findings, Vela laser en-bloc resection approach has a higher muscle inclusion rate for both cT1 and cT2 bladder tumor, compared with CTURBT. (Table 2)
  5. maybe you can also highlight the fact that with more DM involvement in the speciment more early reTURB can be avoided (as a second early invasive procedure with potential compliations, higher costs etc)
    Thank you for your comments. We’ve highlighted this point. (line 19-20 on page 6, Discussion).
  6. Conclusions: Based on presented retrospective data prospective randomized trials are advocated.
    Thank you for your comments. We are going to design the prospective randomized trials on ERBT.

Sincerely yours,

Che-Wei Chang, Tsz-Yi Tang and Yung-Chin Lee

Aug. 24, 2022

Reviewer 2 Report (New Reviewer)

Congratulations for this article. Comprehensive and complete in every field.

The number of patients is valid for comparision.

Regarding surgeon's experience, how long has the surgeon experienced the en bloc resection? How many surgeons practised these 100 bladder cancer? Were they the same beetween the 2 groups?

In your work  you find a Recurrence-free survival (month) of  10.54 for TURBT and 12.01 for en bloc. When did you practice the cystoscopy follow up? You can mention this work of 2016 (10.1016/j.urology.2016.01.004) where  they found a recurrence-free survival of 85% after 2 years and and all the en bloc resection of bladder tumor samples showed the presence of detrusor muscle and the recurrence rate at the first follow-up cystoscopy (3 months) was 5.4%).

What about the re-resection? Did you experience ones? Would you use en bloc again? We suggest you this work ( quote10.1007/s00345-019-02805-8)

What about the follow up? Did you plan a longer one? As you know very few studies investigated long-term oncological outcomes of en bloc resection. Cite this 10.1016/j.euros.2021.01.015 (5 years follow up)

Author Response

Dear editors and reviewers:

Thank you very much for your comments on our manuscript. We have carefully considered the comments of the reviewers and revised our manuscript accordingly.   Below are our responses to the reviewers' comments.

*Our previous manuscript ID: jcm- 1769402

Reviewer 2

  1. Regarding surgeon's experience, how long has the surgeon experienced the en bloc resection?

Our team has experienced this technique for 3 to 4 years.

  1. How many surgeons practised these 100 bladder cancer?
    We have 3 surgeons (Jhen-Hao Jhan, Hsun-Shuan Wang, and Yung-Chin Lee) that practiced this en-bloc resection technique.
  2. Were they the same beetween the 2 groups?
    Thank you for your comments. Yes, they were the same between the 2 groups.
  3. In your work  you find a Recurrence-free survival (month) of  54 for TURBT and 12.01 for en bloc. When did you practice the cystoscopy follow up? You can mention this work of 2016 (10.1016/j.urology.2016.01.004) where  they found a recurrence-free survival of 85% after 2 years and and all the en bloc resection of bladder tumor samples showed the presence of detrusor muscle and the recurrence rate at the first follow-up cystoscopy (3 months) was 5.4%).
    Thank you for your comments. We practice cystoscopy follow-up every 3 months after surgery based on NCCN guideline. We’ve cited this article and added their results in our discussion accordingly. (line 34-37 on page 6)
  4. What about the re-resection? Did you experience ones? Would you use en bloc again? We suggest you this work ( quote10.1007/s00345-019-02805-8)
    Thank you for your comments. No patient received early repeat resection in LERBT group. However, we prefer laser en-bloc resection again if tumor recurrence was detected during follow up. We’ve cited the article and revised the paragraph (line 21-24 on page 6)

  1. What about the follow up? Did you plan a longer one? As you know very few studies investigated long-term oncological outcomes of en bloc resection. Cite this 10.1016/j.euros.2021.01.015 (5 years follow up
    Thank you for your comments. Yes, we will keep track our data for longer follow-up duration. We also cited the article and revised the paragraph in our manuscript (line 37-39 on page 7)   

Sincerely yours,

Che-Wei Chang, Tsz-Yi Tang and Yung-Chin Lee

Aug. 24, 2022

Reviewer 3 Report (New Reviewer)

The safety and efficacy of en-bloc endoscopic resection of 2 bladder tumor with Vela laser

Dear colleagues,

It was my great pleasure to review your manuscript and it my congrats to your nice research paper.

Title:

The title does not match with the methods of the study. You had 2 groups and you compared Laser to conventional TURB.

Abstract:

Good

CTURBT : What C stands for? Conventional ?

Introduction:

Very lengthy, please shorten and highlight the gaps in literature

The conventional transurethral resection of bladder tumor (CTURBT) with monopolar or bipolar resectoscope obtains inferior quality of specimen than laser en-bloc resection 46 of bladder tumor (LERBT) based on previous meta-analysis study [11]. ……..This statement does not give a rational to your study, try to put a gap or a limitation.

Although several studies have been evaluated the thullium laser and holmium laser  en-bloc resection, most of the studies focus on non-muscle invasive bladder tumor. ……you used although, and I could not find any contrary between the two statements in the sentence.

Methods:

Regarding to the intravesical chemotherapies, single shot or 6-8 cycles of instillation of mitomycin C 30mg were performed in selected patients……. please delete a single shot

Why you did not put incidence of perforation as one of the outcome?

Pearson’s chi-square test and student t test for all clinical variables ….. chi-square is for categorical and t-test for numeric variables.

Results:

The mean ages among LERBT and CTURBT are 69.2 and 68.0 years, respectively…if you put the mean, it is recommended to include also the SD.

Table 1:

Including variables that have less incidence like DM, ESRD, and Dyslipidemia makes the analysis is less accurate. You can enumerate the most frequent medical diseases, then to add less common in others.

Tumor number: after single and multiple, I could not understand why you put mean…..Please review!

Please review the data on Mean size (cm), you put the mean with no SD, and why for percentage?

Table 2:

 47.86 ± 2.625 ….please put only 1 decimal and refer to that on the footnotes of the table

2 (4.0%) (one bleeding*, one sepsis)….that can be added to the footnotes of the table

I expected to mention the number of bladder perforation in each arm.

Discussion:

We did not observe obturator nerve reflex in both groups …. You did not mention that in the results nor in tables.

In one study, LERBT showed 12.1% absence of detrusor muscle in specimen while CTURBT showed 6.3%, though there were no statistical differences (p=0.088) [30]……I could not understand this sentence because your results referred to a difference. Then, if you mention one of your results, why you added a reference [30].

Computed tomography urogram (CTU) is the most useful image tool for bladder cancer patients…..I am not sure about this fact, but MRI gives a better staging for bladder cancer. If that, please add a reference.

Conclusion:

No significant differences were noted in terms of peri-operative and oncological outcomes…… please add to monopolar / bipolar TURB

Author Response

Dear editors and reviewers:

Thank you very much for your comments on our manuscript. We have carefully considered the comments of the reviewers and revised our manuscript accordingly.   Below are our responses to the reviewers' comments.

*Our previous manuscript ID: jcm- 1769402

Reviewer 3

  1. Title: The title does not match with the methods of the study. You had 2 groups and you compared Laser to conventional TURB.
    Thank you for your advice. We’ve revised our title.
  2. Abstract:Good; CTURBT : What C stands for? Conventional ?
    Thank you for your kind advice. Yes, we’ve revised it in our manuscript. (line 15-16 on page 1)
  3. Introduction: Very lengthy, please shorten and highlight the gaps in literature

The conventional transurethral resection of bladder tumor (CTURBT) with monopolar or bipolar resectoscope obtains inferior quality of specimen than laser en-bloc resection 46 of bladder tumor (LERBT) based on previous meta-analysis study [11]. ……..This statement does not give a rational to your study, try to put a gap or a limitation.

Although several studies have been evaluated the thullium laser and holmium laser  en-bloc resection, most of the studies focus on non-muscle invasive bladder tumor. ……you used although, and I could not find any contrary between the two statements in the sentence.

Thank you for your comments. We’ve extensively revised to shorten the introduction. (Introduction section)

Methods:

  1. Regarding to the intravesical chemotherapies, single shot or 6-8 cycles of instillation of mitomycin C 30mg were performed in selected patients……. please delete a single shot
    Thank you for your advice. We’ve deleted “a single shot”. (line 36 on page 2)
  2. Why you did not put incidence of perforation as one of the outcome?
    Thank you for your advice. We’ve put the incidence of perforation in our Table 2.
  3. Pearson’s chi-square test and student t test for all clinical variables ….. chi-square is for categorical and t-test for numeric variables.
    Thank you for your advice. We’ve revised the paragraph as your per request. (line 1-2 on page 3)

Results:

  1. The mean ages among LERBT and CTURBT are 69.2 and 68.0 years, respectively…if you put the mean, it is recommended to include also the SD.
    Thank you for your advice. We’ve added SD in our result section. (line 13 on page 3)

Table 1:

  1. Including variables that have less incidence like DM, ESRD, and Dyslipidemia makes the analysis is less accurate. You can enumerate the most frequent medical diseases, then to add less common in others.
    Thank you for your advice. We have removed the 2 variables (ESRD and Dyslipidemia) with less incidence (<10%) and added CKD and smoking status in Table 1.
  2. Tumor number: after single and multiple, I could not understand why you put mean…..Please review!
    Thank you for your advice. We’ve deleted the mean of tumor number. (Table 1)
  3. Please review the data on Mean size (cm), you put the mean with no SD, and why for percentage?
    Thank you for your advice. The data (0.64 and 0.63) is SD. We’ve revised the table 1. Thanks again for your comment.

Table 2:

  1. 86 ± 2.625 ….please put only 1 decimal and refer to that on the footnotes of the table
    Thank you for your advice. We’ve revised the table 2.
  2. 2 (4.0%) (one bleeding*, one sepsis)….that can be added to the footnotes of the table
    Thank you for your advice. We’ve revised the table 2.
  3. I expected to mention the number of bladder perforation in each arm.
    Thank you for your advice. We’ve added the number of bladder perforation in the table 2.

Discussion:

  1. We did not observe obturator nerve reflex in both groups …. You did not mention that in the results nor in tables.
    Thank you for your advice. No obturator reflex jerk was recorded in the medical records that have been added in the result section. (line 26 on page 3)

  1. In one study, LERBT showed 12.1% absence of detrusor muscle in specimen while CTURBT showed 6.3%, though there were no statistical differences (p=0.088) [30]……I could not understand this sentence because your results referred to a difference. Then, if you mention one of your results, why you added a reference [30].
    “In one study, LERBT showed 12.1% absence of detrusor muscle in specimen while CTURBT showed 6.3%, though there were no statistical differences (p=0.088) [30]……” We consider those results were conflicting due to heterogeneity of tumor demography and technologies. For better interpretation and reading, we’ve revised this paragraph. (line 11-18 on page 6)
  2. Computed tomography urogram (CTU) is the most useful image tool for bladder cancer patients…..I am not sure about this fact, but MRI gives a better staging for bladder cancer. If that, please add a reference.
    Thank you for your advice. We’ve revised this sentence. (line 41 on page 6)

Conclusion:

  1. No significant differences were noted in terms of peri-operative and oncological outcomes…… please add to monopolar / bipolar TURB
    Thank you for your advice. We’ve rewritten the paragraph (line 4 on page 7)

Sincerely yours,

Che-Wei Chang, Tsz-Yi Tang and Yung-Chin Lee

Aug. 24, 2022

This manuscript is a resubmission of an earlier submission. The following is a list of the peer review reports and author responses from that submission.

Round 1

Reviewer 1 Report

This is a retrospective comparative clinical study with highly relevant topic of en bloc laser TURB. As such, it bears significant limitations and biases which must be included in the text.

1. operator bias - quality of resection lies in experience

2. selection bias = smaller, better demarkated and better localized tumours are usually picked for en bloc

Complications = only 2 complication occured in one of the 2 groups. This sounds like a bias of small numbers

The main objective - number of DM in the specimen. It would be of interest to mention whether this translated to fewer early reTURB and what the results of those re-resections were (remanent tumours, upstage to T2) because this affects prognosis

My most important objective is the absolute lack of any oncological and histological data post TURB and during the follow-up.

1. Stages, grades, risk groups (recurrence, progression), number of single-shot chemotherapy in either groups, adjuvant instillations, BCG (avarage N of cycles)

Based on these data, risk of recurrence can be estimated using KM curves, but - and this critical - adjusted for all the above mentioned variables within the particular risk groups for recurrence. As such, the numbers of paticipants and lenght of follow-up are insufficient to draw conclusions whether a type of resection itself correlates with RFS.

Author Response

Dear reviewer,

Reviewer 2 Report

Numerous studies have compared en-bloc resection of bladder tumor with conventional TURBT and most of this research's results are similar to previous studies. But only a few of them used vela laser. I have several suggestions or problems:

1. What's the difference between Vela laser and other lasers that used for ERBT? What're its advantages and disadvantages compared with other lasers?

2. What's the criteria for removing the catheter and discharge?

3. The inclusion and exclusion criteria should be more clear.

4. Line 106 "For cT2 tumors, the detrusor muscle inclusion rate is 100% compared to CTURBT group." What's the rate of CTURBT group? 71.4%?

5. Line 56 "Hence, we included all patients with clinical stage cTa, cT1, cTis and cT2 based on CT image" How to differentiate cTa, cT1 or cTis based on CT and the data of clinical stage should be provided in table 1.

6. I think this manuscript should be evaluated by a statistician. 

Reviewer 3 Report

Dear authors, please find my comments below:

What is a high recurrence rate? (please specify) Referred to: “In Taiwan, bladder cancer is the second most common urological cancer with high recurrence rate”

Thallium should be replaced by Thulium laser.

“It is difficult to differentiate non-muscle invasive and muscle invasive bladder tumor solely based on image”. According to new VI-RADS classification this is feasible, please mention in the introduction or discuss.

A standard measure of performance status is not used, such as ECOG status among others.

The process to choose any of the treatment intentions is nebulous. It was surgeon dependent? Aleatory?

The instillation after TURB is underused and is a standard of care. This can impact the results.

Progression free survival is not reported.

More single tumours are seen in the laser arm, moreover, is statistically significant. This is a not well-balanced population. The differences on shorter catheterization time, shorter hospitalization, complications, and recurrence free survival can be explained due these differences.

If your aim is only to see the feasibility of the laser, is not needed to compare with the standard of care, if your aim is to show that is “at least as good” or even better than the standard of care, both arms should be balanced, if not is an “unfair” comparison.

Round 2

Reviewer 1 Report

Thank you for reflecting my comments and suggestions

Author Response

Thank you so much for reviewing the article.

Reviewer 2 Report

I appreciate the authors' efforts in revising the manuscript. However, I don't think the authors has well addressed the comments 5 and 6. In addition, as the table 2 shows, one patient expericenced bleeding complication, then what's the criteria or definition of the bleeding complication?
